# Impact of tonsillectomy on irritable bowel syndrome: A nationwide population-based cohort study

Meng-Che Wu[1], Kevin Sheng-Kai Ma[2,3], Yu-Hsun Wang[4], James Cheng-Chung Wei[5,6,7,8,9]*

1 Division of Gastroenterology, Children's Medical Center, Taichung Veterans General Hospital, Taichung, Taiwan, 2 Department of Dentistry, Chung Shan Medical University, Taichung, Taiwan, 3 Department of Life Science, National Taiwan University, Taipei, Taiwan, 4 Department of Medical Research, Chung Shan Medical University Hospital, Taichung, Taiwan, 5 Department of Rheumatology, BenQ Medical Center, The Affiliated BenQ Hospital of Nanjing Medical University, Nanjing, China, 6 Beijing Tsinghua Changgung Hospital, School of Clinical Medicine, Tsinghua University, Beijing, China, 7 Division of Allergy, Immunology and Rheumatology, Chung Shan Medical University Hospital, Taichung, Taiwan, 8 Institute of Medicine, College of Medicine, Chung Shan Medical University, Taichung, Taiwan, 9 Graduate Institute of Integrated Medicine, China Medical University, Taichung, Taiwan

* jccwei@gmail.com

**Data Availability Statement:** This retrospective cohort study was conducted using data from the National Health Insurance Research Database (NHIRD), a database covering 99% of Taiwan's population of 23 million beneficiaries. The database

## Abstract

Tonsillectomy is a commonly performed surgical procedure worldwide; however, the possible long-term consequences have not been fully explored. The tonsils are secondary lymphoid tissue providing a defensive barrier against pathogens. There are few data in the literature on the relationship between prior tonsillectomy and the risk of irritable bowel syndrome (IBS). The aim of this study was to evaluate the risk of developing IBS in patients who underwent tonsillectomy using a nationwide longitudinal population-based cohort. We identified 1300 patients with tonsillectomy and 2600 matched controls from the Longitudinal Health Insurance Research Database of the National Health Insurance Research Database in Taiwan. Propensity score analysis was used for matching age, gender, comorbidities, medications, and index year at a ratio of 1:2. Multiple Cox regression analysis was used to estimate the adjusted hazard ratio of IBS. Furthermore, sensitivity tests and a stratified analysis were conducted. The incidence of IBS was 3 per 1,000 person-years in tonsillectomy patients, which was higher than the rate of 1.8 per 1,000 person-years found in non-tonsillectomy patients. After adjustment for patients' age, gender, comorbidities, and medications, patients who underwent tonsillectomy had a 1.84-fold risk of developing IBS compared to those without tonsillectomy (adjusted hazard ratio [HR]:1.84; 95% CI, 1.09–3.09). Stratified analysis revealed that a higher adjusted HR of 3.79 (95% CI, 1.35–10.64) in patients ≥50 years old, and there was a marginally significantly higher adjusted HR of 1.98 (95% CI, 0.99–3.95) of developing IBS when the follow-up period was longer than 3 years. Tonsillectomy was associated with a higher risk of developing IBS. Physicians should provide appropriate monitoring of IBS in patients undergoing tonsillectomy.

included all insurance claims data, such as outpatient visits, emergency, and hospitalization. Among them, one million subjects (Longitudinal Health Insurance Research Database) [17–19] were randomly sampled from the 23 million beneficiaries, providing data from 1999 to 2013. The sampled database was de-identified and this study was approved by the Institutional Review Board of Chung Shan Medical University Hospital (Approval number CS15134). All relevant data are within the manuscript.

**Funding:** The author(s) received no specific funding for this work.

**Competing interests:** The authors have declared that no competing interests exist.

## Introduction

Tonsillectomy is one of the most frequently performed surgical interventions worldwide. However, studies have shown that the tonsils are important part of the immune system, playing special roles in pathogen detection and defense. The tonsils form a Waldeyer's ring around the apex of the respiratory and alimentary tracts, providing an early warning of airborne or ingested pathogens/antigens. The tonsils act as gatekeepers and can directly and indirectly resist pathogens by stimulating other immune responses [1, 2]. Previous studies have found a dramatic decline in pre-existing antigen-specific IgA antibodies in the nasopharynx after tonsillectomy [3], suggesting that the tonsils are critical for generating mucosal immunity. Furthermore, the tonsils are also considered to be a reservoir of T cells in addition to the thymus [4]. Changes in immune function after tonsillectomy have been reported, but the results remain controversial. Studies have examined the association between antecedent removal of tonsils and the risk of respiratory disorders [5], infectious diseases [5], asthma [5], premature acute myocardial infarction [6], autoimmune diseases [7], inflammatory bowel disease [8, 9], and cancers [10]. The absence of tonsils might have a lasting impact on human health.

Irritable bowel syndrome (IBS) is a functional gastrointestinal disorder that is characterized by recurrent abdominal pain in association with a change in bowel habit, with diarrhea, constipation, or both [11]. IBS has a considerable negative impact on patients' quality of life and work productivity and is still a mysterious cause of significant distress and morbidity. Impairment of health-related quality of life in IBS patients has been shown to be comparable to, or possibly even more severe than, other serious chronic organic diseases like inflammatory bowel disease, diabetes, hypertension, and end-stage renal failure [12]. The pathophysiology of IBS is multifactorial, including changes in visceral hypersensitivity, intestinal motility, gut-brain interaction, mucosal immunity, and so on. However, many studies showed that alteration of the gut microbiome occurred in the majority of IBS patients [13]. One recent study showed that severity of IBS symptoms was associated with reduced diversity and richness of the intestinal microbiome [14]. There is also evidence implicating antecedent infections of bacteria, viruses, protozoa, and even fungi in the pathogenesis of IBS. This type of IBS is termed post-infectious IBS. To date, the causative factors of IBS remain incompletely understood.

The short-term complications of tonsillectomy are well -explored, whereas relatively little is known about the long-term complications. The absence of tonsils might cause undetectable pathogens to penetrate the alimentary tract, leading to post-infectious IBS [15] or small bowel bacterial overgrowth. It has been postulated that in susceptible populations, acute or recurrent infections increase intestinal permeability by weakening tight junctions [11]. Localized inflammation then develops, followed by subsequent immune activation and even gut dysbiosis [11, 16], which can lead to IBS. The aim of this study was to evaluate the risk of developing IBS after receiving tonsillectomy. We thus hypothesized that tonsillectomy could modify the risk of IBS. This hypothesis was tested by analyzing a nationwide population-based retrospective cohort from Taiwan's National Health Insurance Research Database (NHIRD).

## Material and methods

### Data source

This retrospective cohort study was conducted using data from the National Health Insurance Research Database (NHIRD), a database covering 99% of Taiwan's population of 23 million beneficiaries. The database included all insurance claims data, such as outpatient visits, emergency, and hospitalization. Among them, one million subjects (Longitudinal Health Insurance

Research Database) [17–19] were randomly sampled from the 23 million beneficiaries, providing data from 1999 to 2013. The sampled database was de-identified and this study was approved by the Institutional Review Board of Chung Shan Medical University Hospital (Approval number CS15134).

## Study group and outcome measurement

The study population included patients with records of newly diagnosed chronic diseases of tonsils and adenoids (ICD-9-CM code 474) or peritonsillar abscess (ICD-9-CM code 475) from 2000 to 2010 for at least 3 outpatient visits or one inpatient admission within 2 years in order to ensure the accuracy of diagnosis. The tonsillectomy group included individuals with records of tonsillectomy procedure within 2 years of diagnosis of chronic diseases of tonsils and adenoids or peritonsillar abscess. The index date of this cohort was set as the date occurring two years after the first diagnosis date. The non-tonsillectomy group included patients without records of tonsillectomy procedure after diagnosis of chronic diseases of tonsils and adenoids or peritonsillar abscess. Individuals whose diagnosis of IBS (ICD-9-CM code 564.1) was prior to the index date were excluded. The outcome was defined as new-onset IBS diagnosis (ICD-9-CM code 564.1) during at least three outpatient visits or one hospital admission. The participants were followed up until the occurrence of IBS before the end of 2013, or withdrawal from the National Health Insurance system.

## Covariates and matching

The baseline characteristics included age, gender, hypertension (ICD-9-CM 401–405), hyperlipidemia (ICD-9-CM 272.0–272.4), chronic liver disease (ICD-9-CM 571), chronic kidney disease (ICD-9-CM 585), diabetes (ICD-9-CM 250), chronic obstructive pulmonary disease (COPD) (ICD-9-CM 491, 492, 496), cancer (ICD-9-CM 140–208), coronary artery disease (ICD-9-CM 410–414), autoimmune disease (ICD-9-CM 710.0, 720.0, 714.0), gastrointestinal (GI) infectious disease (ICD-9-CM 001–009), anxiety (ICD-9-CM 300.0), and depression (ICD-9-CM 296.2, 296.3, 300.4 and 311). The aforementioned comorbidities were included in the analyses if they were noted one year before the index date for at least once hospitalization or three outpatient visits. In addition, usage of oral or injected corticosteroids, non-steroidal anti-inflammatory drugs (NSAIDs), opioids, and antibiotics for ≥30 days during the study period were also included in the analyses.

Then, propensity scores were matched by age, gender, hypertension, hyperlipidemia, chronic liver disease, chronic kidney disease, diabetes, chronic obstructive pulmonary disease, cancer, cardiovascular disease, autoimmune disease, corticosteroids, and index year to control for potential confounding factors. The propensity score was a probability estimated by logistic regression, with the binary variable being whether or not patients had received tonsillectomy, i.e., tonsillectomy vs. non-tonsillectomy groups. By matching propensity scores, the heterogeneity of baseline characteristics, and comorbidities between the two groups were balanced.

## Statistical analysis

The comparison of the tonsillectomy group and non-tonsillectomy group was defined as the absolute standardized differences (ASD). The ASD between the covariates of these two groups after propensity score matching were <10%, which indicates a good balance between tonsillectomy and non-tonsillectomy cohorts. Kaplan-Meier analysis was used to calculate the cumulative incidence of IBS and the log-rank test was used to test the significance. Cox proportional hazard model was used to estimate the hazard ratio of IBS between the tonsillectomy group

and non-tonsillectomy group. SPSS version 18.0 (SPSS Inc., Chicago, IL, USA) was used for the statistical analyses.

## Results

### Basic demographic characteristics of the study subjects

A total of 9,410 patients diagnosed with chronic disease of tonsils and adenoids or peritonsillar abscess were identified from the NHI database. Among them, 7,885 patients did not receive tonsillectomy, whereas 1,457 patients received tonsillectomy within two years. After excluding patients diagnosed with IBS before tonsillectomy, there were 6,985 non-tonsillectomy and 1,306 tonsillectomy patients. After propensity score matching, there were 2,600 patients in the non-tonsillectomy group and 1,300 patients in the tonsillectomy group (Fig 1 and Table 1).

### Risk of IBS in tonsillectomy

A total of 59 cases of IBS were identified over 27,005 observed person-years, and 27 of these cases were associated with tonsillectomy. Cumulative probability of IBS was found to be significantly higher for patients with tonsillectomy over a 12-year period of follow-up (Log-rank test, p = 0.039) (Fig 2). The incidence density of IBS patients with tonsillectomy was higher than that among the non-tonsillectomy cohort (3.0 vs 1.8 per 1000 person-years). The Cox proportional hazard model revealed that patients with tonsillectomy had a significantly higher risk of IBS compared to patients without tonsillectomy (adjusted hazard ratio [HR]:1.84, 95%

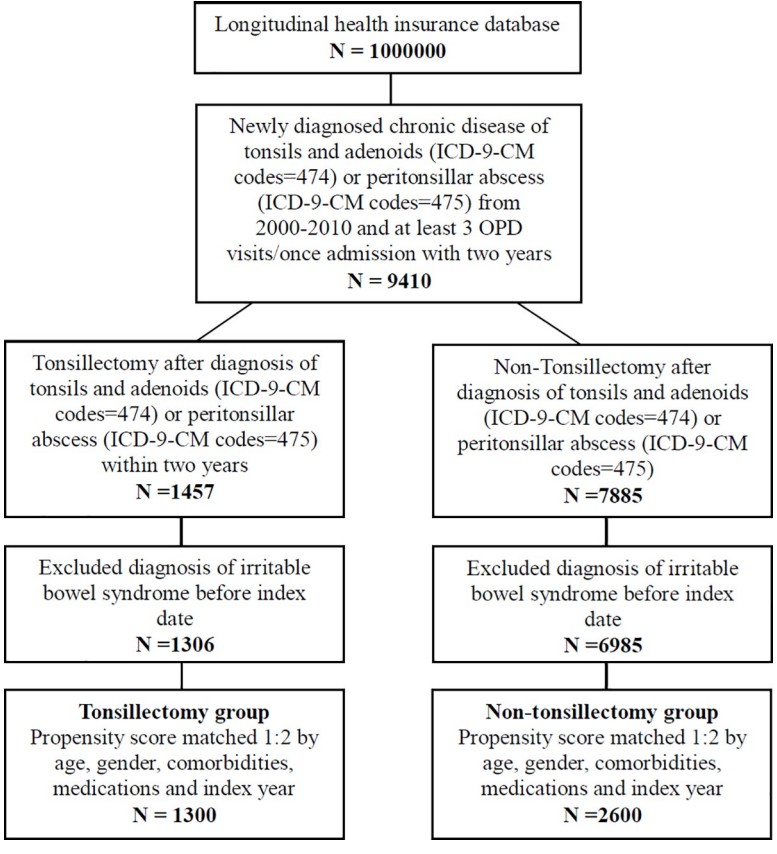

**Fig 1. Flowchart of study.**

**Table 1. Demographic characteristics of tonsillectomy group and non-tonsillectomy group.**

| | Before propensity score matching | | | | | After propensity score matching | | | | |
| | Tonsillectomy (N = 1306) | | Non-Tonsillectomy (N = 6985) | | | Tonsillectomy (N = 1300) | | Non-Tonsillectomy (N = 2600) | | |
| | n | % | n | % | ASD | n | % | n | % | ASD |
|---|---|---|---|---|---|---|---|---|---|---|
| Age | | | | | 0.290 | | | | | 0.067 |
| <20 | 505 | 38.7 | 2223 | 31.8 | | 502 | 38.6 | 1004 | 38.6 | |
| 20–50 | 670 | 51.3 | 3375 | 48.3 | | 667 | 51.3 | 1281 | 49.3 | |
| ≧50 | 131 | 10.0 | 1387 | 19.9 | | 131 | 10.1 | 315 | 12.1 | |
| Mean ± SD | 26.9 ± 16.2 | | 32.2 ± 19.9 | | 0.291 | 26.9 ± 16.2 | | 28 ± 17.3 | | 0.052 |
| Gender | | | | | 0.178 | | | | | 0.003 |
| Female | 513 | 39.3 | 3357 | 48.1 | | 513 | 39.5 | 1022 | 39.3 | |
| Male | 793 | 60.7 | 3628 | 51.9 | | 787 | 60.5 | 1578 | 60.7 | |
| Hypertension | 69 | 5.3 | 574 | 8.2 | 0.117 | 69 | 5.3 | 152 | 5.8 | 0.023 |
| Hyperlipidemia | 43 | 3.3 | 240 | 3.4 | 0.008 | 40 | 3.1 | 61 | 2.3 | 0.045 |
| Chronic liver disease | 36 | 2.8 | 124 | 1.8 | 0.066 | 33 | 2.5 | 63 | 2.4 | 0.007 |
| Chronic kidney disease | 3 | 0.2 | 23 | 0.3 | 0.019 | 3 | 0.2 | 6 | 0.2 | 0.000 |
| Diabetes | 34 | 2.6 | 255 | 3.7 | 0.060 | 34 | 2.6 | 54 | 2.1 | 0.036 |
| COPD | 11 | 0.8 | 114 | 1.6 | 0.071 | 10 | 0.8 | 16 | 0.6 | 0.019 |
| Cancer | 12 | 0.9 | 94 | 1.3 | 0.040 | 12 | 0.9 | 16 | 0.6 | 0.035 |
| Cardiovascular disease | 19 | 1.5 | 149 | 2.1 | 0.051 | 19 | 1.5 | 26 | 1.0 | 0.042 |
| Autoimmune disease | 4 | 0.3 | 29 | 0.4 | 0.018 | 4 | 0.3 | 9 | 0.3 | 0.007 |
| GI infectious disease | 12 | 0.9 | 90 | 1.3 | 0.035 | 12 | 0.9 | 30 | 1.2 | 0.023 |
| GERD | 14 | 1.1 | 52 | 0.7 | 0.035 | 14 | 1.1 | 19 | 0.7 | 0.037 |
| Anxiety | 31 | 2.4 | 125 | 1.8 | 0.041 | 31 | 2.4 | 28 | 1.1 | 0.100 |
| Depression | 25 | 1.9 | 89 | 1.3 | 0.051 | 25 | 1.9 | 36 | 1.4 | 0.042 |
| Corticosteroids | 203 | 15.5 | 1247 | 17.9 | 0.062 | 202 | 15.5 | 401 | 15.4 | 0.003 |
| NSAIDs | 871 | 66.7 | 4818 | 69.0 | 0.049 | 869 | 66.8 | 1666 | 64.1 | 0.058 |
| Antibiotics | 789 | 60.4 | 4343 | 62.2 | 0.036 | 787 | 60.5 | 1477 | 56.8 | 0.076 |
| Opioids | 74 | 5.7 | 387 | 5.5 | 0.005 | 73 | 5.6 | 95 | 3.7 | 0.093 |
| Index year | | | | | 0.360 | | | | | 0.049 |
| 2002 | 135 | 10.3 | 1224 | 17.5 | | 135 | 10.4 | 270 | 10.4 | |
| 2003 | 156 | 11.9 | 966 | 13.8 | | 156 | 12.0 | 313 | 12.0 | |
| 2004 | 149 | 11.4 | 921 | 13.2 | | 149 | 11.5 | 303 | 11.7 | |
| 2005 | 111 | 8.5 | 708 | 10.1 | | 111 | 8.5 | 228 | 8.8 | |
| 2006 | 143 | 10.9 | 590 | 8.4 | | 143 | 11.0 | 293 | 11.3 | |
| 2007 | 113 | 8.7 | 598 | 8.6 | | 113 | 8.7 | 228 | 8.8 | |
| 2008 | 120 | 9.2 | 482 | 6.9 | | 119 | 9.2 | 226 | 8.7 | |
| 2009 | 107 | 8.2 | 450 | 6.4 | | 106 | 8.2 | 213 | 8.2 | |
| 2010 | 91 | 7.0 | 385 | 5.5 | | 91 | 7.0 | 178 | 6.8 | |
| 2011 | 97 | 7.4 | 405 | 5.8 | | 96 | 7.4 | 186 | 7.2 | |
| 2012 | 84 | 6.4 | 256 | 3.7 | | 81 | 6.2 | 162 | 6.2 | |

ASD: absolute Standardized Differences. The ASD between the covariates of these two groups after propensity score matching were less than 0.1, which indicates a good balance between tonsillectomy and non-tonsillectomy cohorts.

COPD: Chronic obstructive pulmonary disease.

GERD: Gastroesophageal reflux disease.

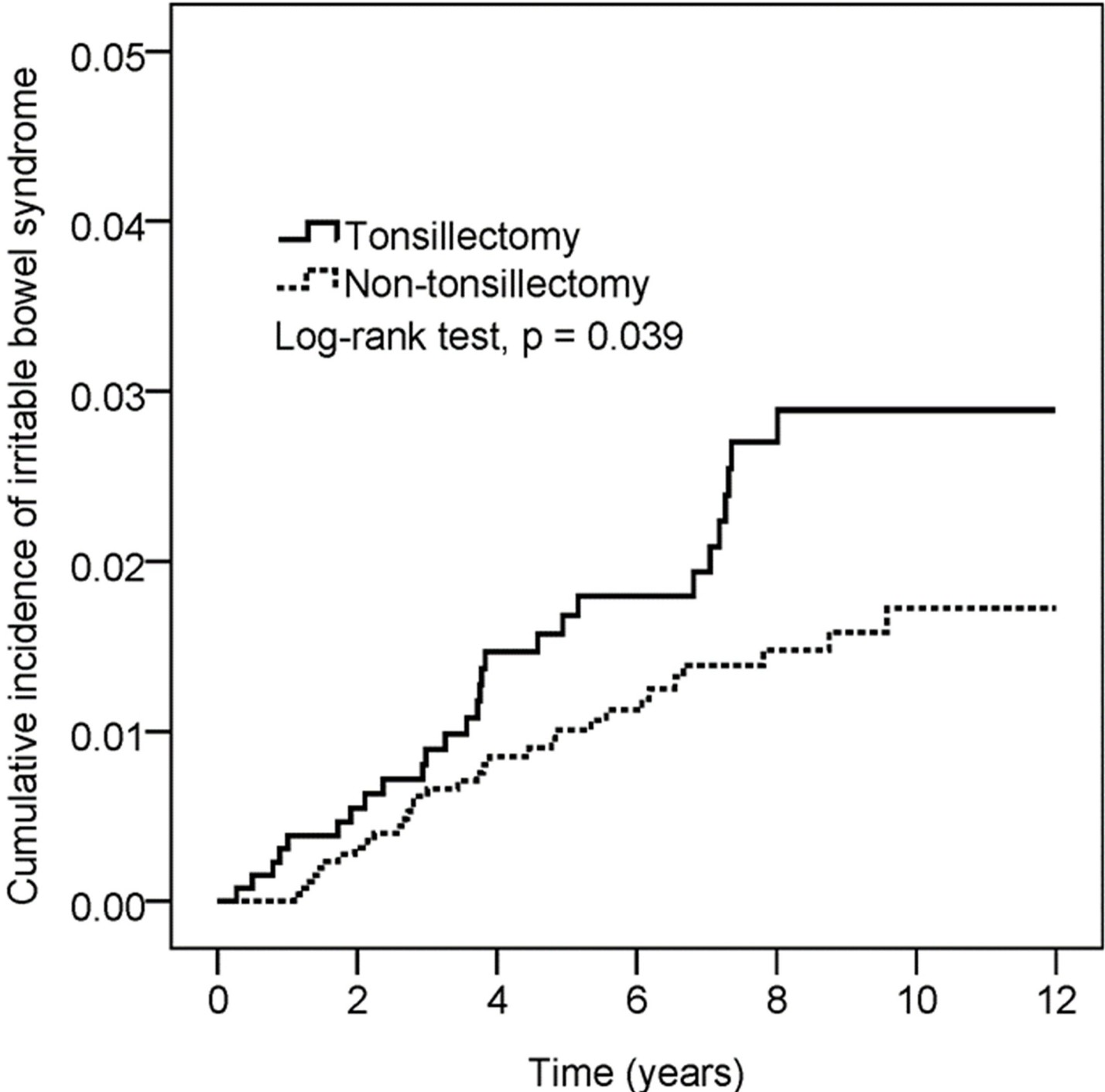

**Fig 2. Kaplan–Meier curves of the cumulative probability of IBS in the study groups.**

CI: 1.09–3.09) after adjusting for possible confounding variables. Patients with hypertension, chronic liver disease, or anxiety had a higher risk of IBS in crude HR, but the adjusted HR showed no significance (adjusted HR: 2.18, 95% CI: 0.90–5.31; adjusted HR 2.41, 95% CI: 0.87–6.73 and adjusted HR: 2.37, 95% CI: 0.61–9.22) (Table 2). The subgroup analysis showed

**Table 2. Multiple Cox proportional hazard regression for the estimation of adjusted hazard ratios for IBS.**

| | No. of IBS | Observed Person-Years | Incidence Density (Per 1000 Person-Years) | Crude HR | 95% C.I. | Adjusted HR[†] | 95% C.I. |
|---|---|---|---|---|---|---|---|
| Group | | | | | | | |
| Non-Tonsillectomy | 32 | 18064 | 1.8 | 1 | | 1 | |
| Tonsillectomy | 27 | 8941 | 3.0 | 1.71 | 1.02–2.85 | 1.84 | 1.09–3.09 |
| Age | | | | | | | |
| <20 | 11 | 11350 | 1.0 | 1 | | 1 | |
| 20–50 | 33 | 12968 | 2.5 | 2.60 | 1.31–5.14 | 2.54 | 1.27–5.09 |
| ≧50 | 15 | 2687 | 5.6 | 5.65 | 2.59–12.32 | 5.85 | 2.41–14.19 |
| Gender | | | | | | | |
| Female | 23 | 10680 | 2.2 | 1 | | 1 | |
| Male | 36 | 16325 | 2.2 | 1.03 | 0.61–1.73 | 1.04 | 0.61–1.8 |
| Hypertension | 9 | 1280 | 7.0 | 3.58 | 1.76–7.29 | 2.18 | 0.90–5.31 |
| Hyperlipidemia | 1 | 511 | 2.0 | 0.86 | 0.12–6.24 | 0.32 | 0.04–2.52 |
| Chronic liver disease | 5 | 608 | 8.2 | 3.91 | 1.56–9.77 | 2.41 | 0.87–6.73 |
| Diabetes | 2 | 457 | 4.4 | 2.00 | 0.49–8.22 | 1.06 | 0.22–5.03 |
| COPD | 1 | 173 | 5.8 | 2.71 | 0.38–19.59 | 1.57 | 0.19–12.85 |
| Cancer | 1 | 209 | 4.8 | 2.33 | 0.32–16.80 | 1.04 | 0.13–8.22 |
| Cardiovascular disease | 1 | 238 | 4.2 | 1.93 | 0.27–13.91 | 0.70 | 0.09–5.84 |
| Autoimmune disease | 1 | 75 | 13.3 | 6.73 | 0.93–48.6 | 5.54 | 0.72–42.72 |
| GI infectious disease | 1 | 328 | 3.0 | 1.41 | 0.20–10.2 | 2.02 | 0.27–14.9 |
| Anxiety | 3 | 368 | 8.1 | 3.85 | 1.21–12.32 | 2.37 | 0.61–9.22 |
| Depression | 2 | 402 | 5.0 | 2.31 | 0.56–9.46 | 1.10 | 0.22–5.53 |
| Corticosteroids | 9 | 4885 | 1.8 | 0.83 | 0.41–1.69 | 0.77 | 0.35–1.70 |
| NSAIDs | 32 | 19726 | 1.6 | 0.44 | 0.26–0.74 | 0.41 | 0.22–0.75 |
| Antibiotics | 31 | 17836 | 1.7 | 0.58 | 0.35–0.97 | 0.74 | 0.40–1.37 |
| Opioids | 3 | 1178 | 2.5 | 1.16 | 0.36–3.70 | 0.74 | 0.21–2.62 |

COPD: Chronic obstructive pulmonary disease.

IBS: Irritable bowel syndrome.

†Adjusted for age, gender, hypertension, hyperlipidemia, chronic liver disease, diabetes, COPD, cancer, cardiovascular disease, autoimmune disease, GI infectious disease, anxiety, depression, corticosteroids, NSAIDs, opioids, and antibiotics.

that there was a difference in the risk of IBS in tonsillectomy patients based on age 50 years or older vs. younger than 50 years (p-value for interaction = 0.214). Specifically, patients aged 50 years or older had a significantly higher risk of developing IBS (adjusted HR: 3.79, 95% CI: 1.35–10.64), whereas there was no significant increase in risk for patients aged between 20 and 50 years, and patients younger than 20 years (Table 3). In addition, there was a marginally significantly higher adjusted HR of 1.98 (95% CI: 0.99–3.95) for developing IBS when the follow-up period was longer than 3 years (Table 4).

## Discussion

In this study, we found a 1.84-fold higher risk of IBS in tonsillectomy patients, and the risk was higher in patients aged ≥ 50 years (adjusted HR of 3.79, 95% CI, 1.35–10.64). Furthermore, there was a marginally significantly higher risk of developing IBS when follow-up was longer than 3 years after tonsillectomy. The association between tonsillectomy and IBS has been explored in one previous case-control study, and prior tonsillectomy was observed in 59.5% and 40.5% of subjects with and without IBS [15]. However, the design of the study might have allowed some information bias to exist and may not have controlled for possible confounding

**Table 3. Sensitivity analysis of adjusted hazard ratios (95% CI) stratified by sex and age groups.**

| | Tonsillectomy | | Non-Tonsillectomy | | | |
|---|---|---|---|---|---|---|
| | N | No. of IBS | N | No. of IBS | HR | 95% C.I. |
| Age | | | | | | |
| <20 | 502 | 5 | 1004 | 6 | 1.71 | 0.52–5.6 |
| 20–50 | 667 | 13 | 1281 | 20 | 1.24 | 0.62–2.49 |
| ≥50 | 131 | 9 | 315 | 6 | 3.79 | 1.35–10.64 |
| | | | | | | p for interaction = 0.214 |
| Gender | | | | | | |
| Female | 513 | 11 | 1022 | 12 | 1.84 | 0.81–4.16 |
| Male | 787 | 16 | 1578 | 20 | 1.62 | 0.84–3.13 |
| | | | | | | p for interaction = 0.808 |

IBS: Irritable bowel syndrome.

factors. To date and to the best of our knowledge, this is the first cohort study to use a nationwide longitudinal population-based dataset to identify an increased IBS risk among patients with prior tonsillectomy. These results indicate that physicians should be alert to the possibility of IBS among patients undergoing tonsillectomy.

We also found that patients diagnosed with hypertension (crude HR, 3.58; 95% CI, 1.76–7.29), chronic liver disease (crude HR, 3.91; 95% CI, 1.56–9.77), or anxiety (crude HR, 3.85; 95% CI, 1.21–12.32) had greater risk of developing IBS. Moreover, patients with autoimmune disease appeared to be marginally related to subsequent IBS in our study. After adjusting for confounders, the adjusted HR of IBS in patients with autoimmune disease was 5.54(95% CI: 0.72–42.72) compared to those without this comorbidity, but this was not statistically significant. There may have been insufficient statistical power to detect significant differences due to the low incidence of autoimmune diseases in our patient population. Although a systematic review including 404 children conducted by Altwairqi et al. concluded that tonsillectomy had no negative impact on both cellular and humoral immunity in children [20], the effect of tonsillectomy on the immunity of adults and the elderly remained unclear. We found that patients aged 50 years or older who underwent tonsillectomy had a 3.79-fold higher risk of IBS. Aging-related immune dysregulation, leading to a chronic inflammatory state and aging-related dysbiosis might be the causes. We suspected that tonsillectomy may worsen pre-existing dysbiosis and immune dysregulation in elder patients and thus increase the risk of IBS.

The pathophysiology underlying the relationship between tonsillectomy and subsequent IBS remains uncertain. Klem et al. reported a meta-analysis of 45 studies that found the pooled risk of developing IBS was 4.2-fold higher in persons who had infectious enteritis in the past

**Table 4. Sensitivity analysis of adjusted hazard ratios (95% CI) stratified by follow-up period.**

| | Tonsillectomy | | Non-Tonsillectomy | | HR | 95% C.I. |
|---|---|---|---|---|---|---|
| | N | No. of IBS | N | No. of IBS | | |
| Follow-up duration | | | | | | |
| ≤3 years[a] | 1300 | 11 | 2600 | 15 | 1.51 | 0.69–3.30 |
| >3 years[b] | 1104 | 16 | 2216 | 17 | 1.98 | 0.99–3.95 |

a: Adjusted for age, gender, hypertension, hyperlipidemia, chronic liver disease, cardiovascular disease, autoimmune disease, and corticosteroids.
b: Adjusted for age, gender, hypertension, chronic liver disease, diabetes, COPD, cancer, and corticosteroids.

year compared with people without infectious enteritis [21]. We speculate that tonsils are important guardians for priming the immune system to pathogens. Tonsils area is the first major mucosa-associated immune barrier that foreign microorganisms have to confront to enter the alimentary tract. The removal of tonsils may allow pathogens to successfully enter the intestine without being detected, altering intestinal flora by influencing bacterial colonization or invasion of the alimentary tracts, leading to post-infectious IBS or even dysbiosis. Andreu-Ballester JC et al. have reported that GALTectomy (such as tonsillectomy and appendectomy) could significantly reduce levels of secretory IgA [3], which was the crucial mucosa-associated antibody against infections. This reduction in patients undergoing tonsillectomy even lasted for more than 20 years in that study [3]. The change of immune defense function may also contribute to the alteration of gut flora in IBS development. Interestingly, we found a higher risk of developing IBS when follow-up was longer than 3 years after tonsillectomy. Our observation implied that it may take a while after tonsillectomy to develop IBS, and repetitive pathogens exposure or altered gut flora over time after tonsillectomy might be the possible explanations. Many studies have demonstrated that dysbiosis of the intestinal microbiome plays important roles in the pathogenesis of IBS [22, 23]. A reproducible finding in IBS patients is a decrease in bacterial diversity and an increase in the temporal instability of the intestinal microbiota [24]. Moreover, a healthy immunity is also an important surveillance system, which helps to control and keep wide variety of microorganisms in the intestinal microbiota in balance [25]. The tonsils are a part of the immune system and comprised mucosa-associated lymphoid tissues that have been reported to have resident CD34+ cells, which contribute to differentiation of several immune cells, including T cells [26]. The association between tonsillectomy and the development of IBS seems immunologically reasonable. Recent research have indicated that CD4+FOXP3+ regulatory T cells (Treg) constitute an important T cell compartment in palatine and lingual tonsils [27] and CD4+CD25+ Treg cells are implicated to play a pivotal role in suppressing intestinal inflammation [28]. It is increasingly clear that low-grade inflammation of gut has been recognized as a key factor in pathophysiology of IBS [29, 30]. Tonsillectomy involving Treg cells depletion in the tonsils might play a role in the development of IBS. Intriguingly, tonsillectomy has been reported to be associated with a higher risk of Behcet's disease and Crohn's disease [9]. These associations might reinforce the proposal that tonsillectomy plays a more important role than was believed in altering gut immunity. Surprisingly, human tonsils were previously proved to be a niche for extrathymic T cell development, thereby contributing to the T cell repertoire, which primarily depends on a functional thymus [31]. Since aging causes a dramatic loss of the thymus, leading to shrinkage in size of the naive T cell repertoire pool, an aging immune system tends to elicit nonspecific inflammation [32]. Tonsils appear to help compensate for this loss by actively maintaining the pool size of naive T cells in older individuals, and by helping to differentiate between commensal and pathogenic bacteria in the alimentary tract to prevent unnecessary immune responses. Therefore, we speculated that tonsillectomy in older patients, would show an especially high correlation with IBS development, compared with their younger counterparts in the present study. Consistent with our finding that there was no significant correlation of tonsillectomy with IBS in younger patients, a recent study found evidences supporting the idea that tonsillectomy has a limited effect on T cell-mediated adaptive immunity in children, indicating that having a functional thymus is rather important in young individuals undergoing this procedure [20]. Nevertheless, the mechanism responsible for tonsil-mediated immune dysfunction, which accelerates IBS development, requires further in-depth study.

The major strengths of this study were the large sample size and the relatively long duration of follow-up, in which a complete history of the medical services used was available for all cases and controls. Therefore, there was minimal selection, information, and recall bias, which

made testing our hypothesis feasible. Furthermore, we used strict exclusion criteria and propensity score matching to control for potential confounders. Nonetheless, there were several limitations that should be noted. Firstly, the NHIRD does not disclose information regarding the patients' diet, socioeconomic status, family history, early adverse life events, personal lifestyle, psychological factors, body weight index, and inflammatory biomarkers, which may be associated risk factors for development of IBS. We used comorbidity variables such as COPD, cardiovascular disease, diabetes and hyperlipidemia as proxy variables to reflect the effects of smoking, dietary preferences, and sociodemographic status. Although we adjusted for various comorbidities and matched propensity scores, these unmeasured confounding factors might have biased our results. Secondly, the diagnoses of chronic disease of tonsils and adenoids, tonsillectomy, and comorbidities were entirely dependent on the ICD-9 codes in the administrative dataset. Therefore, validation of the accuracy of diagnoses could not be verified by personal review of medical records and may have resulted in misclassification. It is worth noting that these misclassifications are more likely to be random, and the associations are often underestimated rather than overestimated. While there is no confirmatory test for IBS, the diagnostic code based on the Rome criteria for IBS are widely used in Taiwan. Most clinicians in Taiwan used the Rome III criteria after 2006 and the Rome II criteria before 2006 for diagnosis of IBS. However, clinical judgment may differ among physicians, so diagnoses may have varied, which could have affected the validity of these diagnoses. However, Taiwan's NHI administration has established an ad hoc committee to monitor the accuracy of claims data to prevent violations. In addition, we only selected subjects that were repeatedly coded to improve the accuracy and validity of the diagnosis. Thirdly, due to the potential for residual confounding inherent in database research, the results should be interpreted with caution. Randomized control trials that are conducted to determine the effect of tonsillectomy and subsequent IBS are laborious and resource intensive. Such studies are difficult to conduct due to ethical issues involved in randomizing patients to undergo tonsillectomy to observe adverse outcomes. Nevertheless, we employed strict exclusion criteria and propensity score matching to control for potential confounders in order to emulate the advantages of a randomized clinical trial. Finally, the majority of the patients were Taiwanese and thus our findings may not be generalizable to other ethnic groups. Clinical studies that include patients from other countries and with different ethnicities should be conducted to confirm our result.

There is growing evidence showing that non-surgical management may be an effective and safe treatment for chronic/recurrent acute tonsillitis in children and adults [33, 34]. Our research supports evidence of an association between prior tonsillectomy and subsequent risk of IBS. Hence, the potential "benefit" of tonsillectomy must be weighed judiciously after considering the personal risk-benefit profile. Furthermore, patients undergoing tonsillectomy should be appropriately monitored for symptoms of IBS.

## Conclusion

Tonsillectomy was associated with a higher risk of developing irritable bowel syndrome in this nationwide, population-based cohort study. Further comprehensive basic and clinical research is warranted to elucidate the mechanisms underlying these associations.

## Acknowledgments

This manuscript was edited by Wallace Academic Editing.

## Author Contributions

**Conceptualization:** Meng-Che Wu.

**Data curation:** Yu-Hsun Wang.

**Formal analysis:** Meng-Che Wu, Yu-Hsun Wang.

**Funding acquisition:** James Cheng-Chung Wei.

**Investigation:** Meng-Che Wu, Kevin Sheng-Kai Ma, Yu-Hsun Wang.

**Methodology:** Yu-Hsun Wang.

**Project administration:** James Cheng-Chung Wei.

**Resources:** Yu-Hsun Wang, James Cheng-Chung Wei.

**Software:** Yu-Hsun Wang.

**Supervision:** James Cheng-Chung Wei.

**Validation:** Yu-Hsun Wang.

**Visualization:** Meng-Che Wu.

**Writing – original draft:** Meng-Che Wu, Kevin Sheng-Kai Ma.

**Writing – review & editing:** Meng-Che Wu, Kevin Sheng-Kai Ma, James Cheng-Chung Wei.

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
