## [Decision Letter · Decision Letter 0]

29 Jun 2020

PONE-D-20-01867

Impact of tonsillectomy on irritable bowel syndrome: A nationwide population-based cohort study

PLOS ONE

Dear Dr. Wei,

Thank you for submitting your manuscript to PLOS ONE. After careful consideration, we feel that it has merit but does not fully meet PLOS ONE’s publication criteria as it currently stands. Therefore, we invite you to submit a revised version of the manuscript that addresses the points raised during the review process.

We look forward to receiving your revised manuscript.

Kind regards,

Sanjiv Mahadeva, MRCP, MD

Academic Editor

PLOS ONE

Additional Editor Comments:

This is an interesting epidemiological association study between tonsillectomy & IBS prevalence. As both reviewers have alluded to - please provide information about history of past GI infection or the use of antibiotics in the cohort studied. This should be included in the multi-variate analysis as well.

Journal Requirements:

2. We note you have included a table to which you do not refer in the text of your manuscript. Please ensure that you refer to Table 1 in your text; if accepted, production will need this reference to link the reader to the Table.

Reviewers' comments:

Reviewer's Responses to Questions

**Comments to the Author**

1. Is the manuscript technically sound, and do the data support the conclusions?

Reviewer #1: Yes

Reviewer #2: Partly

2. Has the statistical analysis been performed appropriately and rigorously? 

Reviewer #1: Yes

Reviewer #2: No

3. Have the authors made all data underlying the findings in their manuscript fully available?

Reviewer #1: Yes

Reviewer #2: Yes

4. Is the manuscript presented in an intelligible fashion and written in standard English?

Reviewer #1: Yes

Reviewer #2: Yes

5. Review Comments to the Author

Reviewer #1: This is an interesting article suggesting that having a tonsillectomy is a risk factor for IBS. The authors do a very good job describing the trial and the data. It is very well written and clear.

I was wondering if the authors examined post-infectious IBS. Food poisoning is a common cause of IBS. Perhap the tonsils are important guardians for priming the immune system to pathogens. Could these two factor be linked. Their database could be used to show this.

Reviewer #2: Thank you to the authors for looking at an interesting part of IBS epidemiology and natural history.

Firstly, what the authors describe is an association, and I would be hesitant to write in their conclusions that patients with tonsillectomy have a significantly higher risk of developing IBS, especially since there are so many factors that are not controlled.

Under covariates and matching paragraph, how did the authors choose the variables to do propensity matching. These are not the usual variables that we look at when we analyse a functional GI cohort. Other more important variables to look at include mental health issues, or GERD or even early adverse life events. I appreciate these are hard data to get from such databases, but therefore, the conclusions cannot be made without controlling for these things.

For Table 1, there is a need to elaborate in the column under ASD, what each number means for the general reader who may not be sure.

The authors found an interesting result of older patients and also those on longer follow up developing IBS. Perhaps they should shed light based on their literature search on possible explanations for these phenomenons.

For their explanations on the reason tonsillectomies predispose patients to IBS, I feel it is overly simplistic to think that tonsils can modulate gut microbiota since the gut immune system distal to the tonsils are complex and more directly involved in controlling pathogens. Are there other factors such as antibiotic use or analgesia used, that the authors feel can mediate this association?

6. PLOS authors have the option to publish the peer review history of their article (what does this mean?). If published, this will include your full peer review and any attached files.

Reviewer #1: No

Reviewer #2: No

---

## [Author Response · Author response to Decision Letter 0]

7 Aug 2020

Dr. Sanjiv Mahadeva, MRCP, MD

Academic Editor, 

Journal: PLOS ONE 

Dear Dr. Sanjiv Mahadeva, 

On behalf of my coauthors, I thank the Reviewers for the comments and am grateful for the opportunity to revise our manuscript (ID: PONE-D-20-01867) titled, “

Impact of tonsillectomy on irritable bowel syndrome: A nationwide population-based cohort study”, for consideration of publication as an Original Research in PLOS ONE. Your comments and those of the reviewers were highly insightful and enabled us to improve the quality of our document. In the following pages are our responses to each comment from the reviewer(s) as well as your own comments. 

Revisions in the text are shown [yellow highlights]. We have addressed the reviewers’ comments point-by-point, and the corresponding changes have been made in track change in the manuscript, which you will find uploaded alongside this document. We hope that our revisions to the document combined with our accompanying responses will be sufficient to render our document suitable for publication in PLOS ONE.

Kind regards,

James Cheng-Chung Wei, MD, PhD

Institute of Medicine, Chung Shan Medical University.

No. 110, Sec 1, Jianguo N. Road, Taichung, 40201, Taiwan

jccwei@gmail.com

Responses to the Additional Editor Comments:

This is an interesting epidemiological association study between tonsillectomy & IBS prevalence. As both reviewers have alluded to - please provide information about history of past GI infection or the use of antibiotics in the cohort studied. This should be included in the multi-variate analysis as well. 

Response: 

 Many thanks you for your comments, we have done and enrolled history of past GI infections and antibiotics usage in the cohort study for multi-variate analysis as well. After multi-variate analysis and adjustment for patients’ age, gender, comorbidities, and medications, patients who underwent tonsillectomy had a 1.84-fold risk of developing IBS compared to those without tonsillectomy (adjusted hazard ratio [HR]:1.84; 95% CI, 1.09-3.09). 

 The adjusted HR of developing IBS did not decrease even after adjusting for GI infectious disease, anxiety, depression, analgesics and antibiotics (according to the suggestions of reviewers’ comments). 

Dear editor, your comments help us to promote quality of this manuscript and we appreciate your kind suggestions.

Responses to the Reviewers’ Comments:

Manuscript ID: PONE-D-20-01867

Manuscript title: Impact of tonsillectomy on irritable bowel syndrome: A nationwide population-based cohort study

Journal: PLOS ONE

Reviewer(s)' Comments to Author:

Reviewer #1: This is an interesting article suggesting that having a tonsillectomy is a risk factor for IBS. The authors do a very good job describing the trial and the data. It is very well written and clear. I was wondering if the authors examined post-infectious IBS. Food poisoning is a common cause of IBS. Perhaps the tonsils are important guardians for priming the immune system to pathogens. Could these two factors be linked. Their database could be used to show this. 

Response: 

 Thank you for your comments. Gastrointestinal (GI) infections and food poisoning were worth exploring. We further enrolled gastrointestinal (GI) infectious disease (ICD-9-CM 001-009), anxiety (ICD-9-CM 300.0), and depression (ICD-9-CM 296.2, 296.3, 300.4 and 311). The aforementioned comorbidities were included in the analyses if they were noted one year before the index date for at least once hospitalization or three outpatient visits. In addition, usage of oral or injected corticosteroids, non-steroidal anti-inflammatory drugs (NSAIDs), opioids, and antibiotics for ≥30 days during the study period were also included and adjusted in the Cox proportional hazard model analyses. (Anxiety, depression, analgesics such as NSAIDs or opioids, antibiotics usage were suggested to examine according to reviewer#2 comments) 

 However, there was no food poisoning event in both cohorts using claimed code: ICD-9-CM external causes of injury codes (E-Codes), which include “unintentional poisoning, such as by foodstuffs (E865) for food poisoning. Therefore, it is not applicable for multi-variate analysis.

p.s. 

The official NHI database employ the ICD-9-CM coding system in Taiwan, so we selected cases by ICD-9-CM external causes of injury codes (E-Codes), which include “unintentional poisoning, such as by foodstuffs (E865) for food poisoning. However, there was no food poisoning event in both cohorts using claimed code: accidental poisoning from unspecified foodstuff (E865). Therefore, it is not applicable for multi-variate analysis. Besides, the code, ICD-9-CM 005 (other bacterial food poisoning), have been enrolled in the cases of GI infections for further multi-variate analysis.

GI infections (ICD-9-CM 001-009) were analyzed. The codes include: 001 (cholera), 002 (typhoid and paratyphoid fever), 003 (salmonellosis), 004 (shigellosis), 005 (other bacterial food poisoning), 006 (amoebiasis), 007 (other protozoan intestinal diseases), 008 (intestinal infections due to other organisms including 00861 enteritis due to rotavirus), 009 (ill-defined intestinal infections)

We revised the paragraph in method section:

“The baseline characteristics included age, gender, hypertension (ICD-9-CM 401-405), hyperlipidemia (ICD-9-CM 272.0-272.4), chronic liver disease (ICD-9-CM 571), chronic kidney disease (ICD-9-CM 585), diabetes (ICD-9-CM 250), chronic obstructive pulmonary disease (COPD) (ICD-9-CM 491, 492, 496), cancer (ICD-9-CM 140-208), coronary artery disease (ICD-9-CM 410-414), autoimmune disease (ICD-9-CM 710.0, 720.0, 714.0), gastrointestinal (GI) infectious disease (ICD-9-CM 001-009), anxiety (ICD-9-CM 300.0), and depression (ICD-9-CM 296.2, 296.3, 300.4 and 311). The aforementioned comorbidities were included in the analyses if they were noted one year before the index date for at least once hospitalization or three outpatient visits. In addition, usage of oral or injected corticosteroids, non-steroidal anti-inflammatory drugs (NSAIDs), opioids, and antibiotics for ≥30 days during the study period were also included in the analyses.” 

 After multi-variate analysis and adjustment for patients’ age, gender, comorbidities, and medications, patients who underwent tonsillectomy had a 1.84-fold risk of developing IBS compared to those without tonsillectomy (adjusted hazard ratio [HR]:1.84; 95% CI, 1.09-3.09). 

 In addition, we further analyzed GI infections (ICD-9-CM 001-009) in observational periods since index date. The results were not significantly different between the tonsillectomy and non-tonsillectomy cohorts. GI infections in observational periods did not show significant influence on IBS risk in our study. 

We also add the sentence as your suggestion in 3rd paragraph of discussion section: 

“………..We speculate that tonsils are important guardians for priming the immune system to pathogens………………..”

Besides, we have added revised figure 1, table 1 and table 2 in the revised manuscript. 

We thank you again for your best comments. Dear reviewer, your comments helps us to promote quality of this manuscript and we appreciate your kind suggestions.

Reviewer #2: Thank you to the authors for looking at an interesting part of IBS epidemiology and natural history. 

1. Firstly, what the authors describe is an association, and I would be hesitant to write in their conclusions that patients with tonsillectomy have a significantly higher risk of developing IBS, especially since there are so many factors that are not controlled.

Under covariates and matching paragraph, how did the authors choose the variables to do propensity matching. These are not the usual variables that we look at when we analyse a functional GI cohort. Other more important variables to look at include mental health issues, or GERD or even early adverse life events. I appreciate these are

hard data to get from such databases, but therefore, the conclusions cannot be made without controlling for these things.

Response: 

 Many thanks for your comments, we revised the conclusion carefully as below: 

“Tonsillectomy was associated with a higher risk of developing irritable bowel syndrome in this nationwide, population-based cohort study. …………” 

 We considered the baseline comorbidities based on the recently published papers (as references below) using the Longitudinal Health Insurance Research Database of the National Health Insurance Research Database in Taiwan to explore the same outcome, ‘irritable bowel syndrome’. Baseline comorbidities used as relevant variables were listed in these references. In addition to the baseline comorbidities used in our research, other more important variables for developing IBS as reviewers’ suggestion were worth exploring, such as mental health issues including anxiety and depression, GERD or even early adverse life events. However, no IBS event was found in patients with the comorbidity of GERD in both cohorts (14 in tonsillectomy and 19 in non-tonsillectomy cohorts with GERD comorbidity after propensity scoring matching, as shown in revised table 1; however, no IBS event found in the study period in both cohorts). Therefore, it is not applicable for multi-variate analysis. Early adverse life events were hard data to get from our database, it is an inherent limitation of population-based datasets such as the NHIRD. We have added this point as a limitation in discussion section. 

 We further enrolled gastrointestinal (GI) infectious disease (ICD-9-CM 001-009), anxiety (ICD-9-CM 300.0), and depression (ICD-9-CM 296.2, 296.3, 300.4 and 311). The aforementioned comorbidities were included in the analyses if they were noted one year before the index date for at least once hospitalization or three outpatient visits. In addition, usage of oral or injected corticosteroids, non-steroidal anti-inflammatory drugs (NSAIDs), opioids, and antibiotics for ≥30 days during the study period were also included and adjusted in the Cox proportional hazard model analyses. (GI infectious disease was suggested to explore according to reviewer#1 comments) 

 After multi-variate analysis and adjustment for patients’ age, gender, comorbidities, and medications, patients who underwent tonsillectomy had a 1.84-fold risk of developing IBS compared to those without tonsillectomy (adjusted hazard ratio [HR]:1.84; 95% CI, 1.09-3.09).

References: 

Lei WY, Chang CY, Wu JH, Lin FH, Hsu Chen C, Chang CF, et al. An Initial Attack of Urinary Stone Disease Is Associated with an Increased Risk of Developing New-Onset Irritable Bowel Syndrome: Nationwide Population-Based Study. PLoS One. 2016;11(6):e0157701. doi: 10.1371/journal.pone.0157701. 

Shen TC, Lin CL, Wei CC, Chen CH, Tu CY, Hsia TC, et al. Bidirectional Association between Asthma and Irritable Bowel Syndrome: Two Population-Based Retrospective Cohort Studies. PLoS One. 2016;11(4):e0153911. doi: 10.1371/journal.pone.0153911.

Liang CM, Hsu CH, Chung CH, Chen CY, Wang LY, Hsu SD, et al. Risk for Irritable Bowel Syndrome in Patients with Helicobacter Pylori Infection: A Nationwide Population-Based Study Cohort Study in Taiwan. Int J Environ Res Public Health. 2020;17(10). doi: 10.3390/ijerph17103737. 

Yang CY, Wu MC, Lin MC, Wei JC. Risk of irritable bowel syndrome in patients who underwent appendectomy: A nationwide population-based cohort study. EClinicalMedicine. 2020;23:100383. doi: 10.1016/j.eclinm.2020.100383. 

We have revised the paragraph in method section:

“The baseline characteristics included age, gender, hypertension (ICD-9-CM 401-405), hyperlipidemia (ICD-9-CM 272.0-272.4), chronic liver disease (ICD-9-CM 571), chronic kidney disease (ICD-9-CM 585), diabetes (ICD-9-CM 250), chronic obstructive pulmonary disease (COPD) (ICD-9-CM 491, 492, 496), cancer (ICD-9-CM 140-208), coronary artery disease (ICD-9-CM 410-414), autoimmune disease (ICD-9-CM 710.0, 720.0, 714.0), gastrointestinal (GI) infectious disease (ICD-9-CM 001-009), anxiety (ICD-9-CM 300.0), and depression (ICD-9-CM 296.2, 296.3, 300.4 and 311). The aforementioned comorbidities were included in the analyses if they were noted one year before the index date for at least once hospitalization or three outpatient visits. In addition, usage of oral or injected corticosteroids, non-steroidal anti-inflammatory drugs (NSAIDs), opioids, and antibiotics for ≥30 days during the study period were also included in the analyses.” 

And, we have made corresponding changes to the results, discussion section, and also in table 1-2.

In 2nd paragraph of results section:

…………………..The Cox proportional hazard model revealed that patients with tonsillectomy had a significantly higher risk of IBS compared to patients without tonsillectomy (adjusted hazard ratio [HR]:1.84, 95% CI: 1.09-3.09) after adjusting for possible confounding variables. Patients with hypertension, chronic liver disease, or anxiety had a higher risk of IBS in crude HR, but the adjusted HR showed no significance (adjusted HR: 2.18, 95% CI: 0.90-5.31; adjusted HR 2.41, 95% CI: 0.87-6.73 and adjusted HR: 2.37, 95% CI: 0.61-9.22) (Table 2). …………

In 2nd paragraph of discussion section:

We also found that patients diagnosed with hypertension (crude HR, 3.58; 95% CI, 1.76-7.29), or chronic liver disease (crude HR, 3.91; 95% CI, 1.56-9.77), or anxiety (crude HR, 3.85; 95% CI, 1.21-12.32) had greater risk of developing IBS. …………. 

We have also revised the 4rd paragraph of discussion section and added ‘early adverse life events’ as a limitation as below:

………………Firstly, the NHIRD does not disclose information regarding the patients’ diet, socioeconomic status, family history, early adverse life events, personal lifestyle, psychological factors, body weight index, and inflammatory biomarkers, which may be associated risk factors for development of IBS. 

2. For Table 1, there is a need to elaborate in the column under ASD, what each number means for the general reader who may not be sure. 

Response: 

Thank you for your suggestion. We add a footnote about ASD in table 1. 

ASD: absolute Standardized Differences. The ASD between the covariates of these two groups after propensity score matching were less than 0.1, which indicates a good balance between tonsillectomy and non-tonsillectomy cohorts.

3. The authors found an interesting result of older patients and also those on longer follow up developing IBS. Perhaps they should shed light based on their literature search on possible explanations for these phenomenons. For their explanations on the reason tonsillectomies predispose patients to IBS, I feel it is overly simplistic to think

that tonsils can modulate gut microbiota since the gut immune system distal to the tonsils are complex and more directly involved in controlling pathogens. 

Response: 

 Thank you for your suggestions. For explanations on the reason tonsillectomies predispose patients to IBS, we have revised the discussion section based on literature search on possible explanations for these phenomenons as below. We have focused on how tonsillectomy can alter immunity (such as by reducing secretory IgA and involving Treg cells depletion) and further strengthened the description of tonsils as important guardians for priming the immune system to pathogens, and added new references to shed light the possible mechanisms underlying these associations. 

In 3rd paragraph of discussion section: 

“…………………We speculate that tonsils are important guardians for priming the immune system to pathogens. Tonsils area is the first major mucosa-associated immune barrier that foreign microorganisms have to confront to enter the alimentary tract. The removal of tonsils may allow pathogens to successfully enter the intestine without being detected, altering intestinal flora by influencing bacterial colonization or invasion of the alimentary tracts, leading to post-infectious IBS or even dysbiosis. Andreu-Ballester JC et al. have reported that GALTectomy (such as tonsillectomy and appendectomy) could significantly reduce levels of secretory IgA [3], which was the crucial mucosa-associated antibody against infections. This reduction in patients undergoing tonsillectomy even lasted for more than 20 years in that study [3]. The change of immune defense function may also contribute to the alteration of gut flora in IBS development. Interestingly, we found a higher risk of developing IBS when follow-up was longer than 3 years after tonsillectomy. Our observation implied that it may take a while after tonsillectomy to develop IBS, and repetitive pathogens exposure or altered gut flora over time after tonsillectomy might be the possible explanations. Many studies have demonstrated that dysbiosis of the intestinal microbiome plays important roles in the pathogenesis of IBS [22, 23]. A reproducible finding in IBS patients is a decrease in bacterial diversity and an increase in the temporal instability of the intestinal microbiota [24]. Moreover, a healthy immunity is also an important surveillance system, which helps to control and keep wide variety of microorganisms in the intestinal microbiota in balance [25]. The tonsils are a part of the immune system and comprised mucosa-associated lymphoid tissues that have been reported to have resident CD34+ cells, which contribute to differentiation of several immune cells, including T cells [26]. The association between tonsillectomy and the development of IBS seems immunologically reasonable. Recent research have indicated that CD4+FOXP3+ regulatory T cells (Treg) constitute an important T cell compartment in palatine and lingual tonsils [27] and CD4+CD25+ Treg cells are implicated to play a pivotal role in suppressing intestinal inflammation [28]. It is increasingly clear that low-grade inflammation of gut has been recognized as a key factor in pathophysiology of IBS [29, 30]. Tonsillectomy involving Treg cells depletion in the tonsils might play a role in the development of IBS. Intriguingly, tonsillectomy has been reported to be associated with a higher risk of Behcet’s disease and Crohn’s disease [9]. These associations might reinforce the proposal that tonsillectomy plays a more important role than was believed in altering gut immunity. ……………………………………………………………………………………………………………………………………………………………Nevertheless, the mechanism responsible for tonsil-mediated immune dysfunction, which accelerates IBS development, requires further in-depth study.”

We have added references in discussion section: 

3. Andreu-Ballester JC, Perez-Griera J, Ballester F, Colomer-Rubio E, Ortiz-Tarin I, Penarroja Otero C. Secretory immunoglobulin A (sIgA) deficiency in serum of patients with GALTectomy (appendectomy and tonsillectomy). Clin Immunol. 2007;123(3): 289-297. doi: 10.1016/j.clim.2007.02.004.

27. Palomares O, Rückert B, Jartti T, Kücüksezer UC, Puhakka T, Gomez E, et al. Induction and maintenance of allergen-specific FOXP3+ Treg cells in human tonsils as potential first-line organs of oral tolerance. J Allergy Clin Immunol. 2012;129(2):510-20, 20.e1-9. doi: 10.1016/j.jaci.2011.09.031. 

28. Makita S, Kanai T, Nemoto Y, Totsuka T, Okamoto R, Tsuchiya K, et al. Intestinal lamina propria retaining CD4+CD25+ regulatory T cells is a suppressive site of intestinal inflammation. J Immunol. 2007;178(8):4937-46. doi: 10.4049/jimmunol.178.8.4937. 

29. Sinagra E, Pompei G, Tomasello G, Cappello F, Morreale GC, Amvrosiadis G, et al. Inflammation in irritable bowel syndrome: Myth or new treatment target? World J Gastroenterol. 2016;22(7):2242-55. doi: 10.3748/wjg.v22.i7.2242. 

30. Akiho H, Ihara E, Nakamura K. Low-grade inflammation plays a pivotal role in gastrointestinal dysfunction in irritable bowel syndrome. World J Gastrointest Pathophysiol. 2010;1(3):97-105. doi: 10.4291/wjgp.v1.i3.97.

 We have mentioned in our discussion section regarding older patients would show a higher correlation with IBS development:

………………………………Surprisingly, human tonsils were previously proved to be a niche for extrathymic T cell development, thereby contributing to the T cell repertoire, which primarily depends on a functional thymus [31]. Since aging causes a dramatic loss of the thymus, leading to shrinkage in size of the naive T cell repertoire pool, an aging immune system tends to elicit nonspecific inflammation [32]. Tonsils appear to help compensate for this loss by actively maintaining the pool size of naive T cells in older individuals, and by helping to differentiate between commensal and pathogenic bacteria in the alimentary tract to prevent unnecessary immune responses. Therefore, we speculated that tonsillectomy in older patients, would show an especially high correlation with IBS development, compared with their younger counterparts in the present study………………………………………………………..

 We also added the description in 2nd paragraph of discussion section regarding the elderly who underwent tonsillectomy had a higher risk of IBS.

“…………….Although a systematic review including 404 children conducted by Altwairqi et al. concluded that tonsillectomy had no negative impact on both cellular and humoral immunity in children [20], the effect of tonsillectomy on the immunity of adults and the elderly remained unclear. We found that patients aged 50 years or older who underwent tonsillectomy had a 3.79-fold higher risk of IBS. Aging-related immune dysregulation, leading to a chronic inflammatory state and aging-related dysbiosis might be the causes. We suspected that tonsillectomy may worsen pre-existing dysbiosis and immune dysregulation in elder patients and thus increase the risk of IBS.”

4. Are there other factors such as antibiotic use or analgesics used, that the

authors feel can mediate this association?

Response: 

 Thank you for your comments. Usage of analgesics such as non-steroidal anti-inflammatory drugs (NSAIDs), opioids, and antibiotics for ≥30 days during the study period were also included in the multi-variate analyses. After multi-variate analysis and adjustment for patients’ age, gender, comorbidities, and medications, patients who underwent tonsillectomy had a 1.84-fold risk of developing IBS compared to those without tonsillectomy (adjusted hazard ratio [HR]:1.84; 95% CI, 1.09-3.09). 

 The adjusted HR of developing IBS did not decrease even after adjusting for analgesics and antibiotics. 

Dear reviewer, I greatly appreciate your time and your kind comments again; your comments reinforced our manuscript and we did our best to respond to your comments carefully.

---

## [Editor Report · Decision Letter 1]

13 Aug 2020

Impact of tonsillectomy on irritable bowel syndrome: A nationwide population-based cohort study

PONE-D-20-01867R1

Dear Dr. Wei,

We’re pleased to inform you that your manuscript has been judged scientifically suitable for publication and will be formally accepted for publication once it meets all outstanding technical requirements.

Kind regards,

Sanjiv Mahadeva, MRCP, MD

Academic Editor

PLOS ONE

Additional Editor Comments (optional):

The authors have made sufficient revision to the original manuscript according to the (minor) comments from the 2 reviewers.

---

## [Editor Report · Acceptance letter]

21 Aug 2020

PONE-D-20-01867R1 

Impact of tonsillectomy on irritable bowel syndrome: A nationwide population-based cohort study 

Dear Dr. Wei:

I'm pleased to inform you that your manuscript has been deemed suitable for publication in PLOS ONE. Congratulations! Your manuscript is now with our production department. 

Kind regards, 

on behalf of

Dr. Sanjiv Mahadeva 

Academic Editor

PLOS ONE